# Efficacy and safety of guide extension catheter in balloon pulmonary angioplasty for treatment of complex lesions in chronic thromboembolic pulmonary hypertension

**Masao Takigami[1], Naohiko Nakanishi[1]\*, Hideo Tsubata[1], Kuniyoshi Fukai[2], Yuki Matsubara[1], Kenji Yanishi[1], Kan Zen[1], Takeshi Nakamura[1], Satoaki Matoba[1]**

1 Department of Cardiovascular Medicine, Graduate School of Medical Science, Kyoto Prefectural University of Medicine, Kyoto, Japan, 2 Department of Cardiovascular Medicine, Omihachiman Community Medical Center, Shiga, Japan

\* naka-nao@koto.kpu-m.ac.jp

## Abstract

### Background

Balloon pulmonary angioplasty (BPA) is used for treatment of inoperable chronic thromboembolic pulmonary hypertension (CTEPH) and residual pulmonary hypertension after pulmonary endarterectomy (PEA) to improve hemodynamics, right ventricular function, and exercise capacity. However, the effectiveness and safety of guide extension catheters for BPA treatment in patients with CTEPH have not been demonstrated.

### Methods

We retrospectively analyzed 91 lesions in 55 sessions of 28 patients with CTEPH who underwent BPA using a guide extension catheter. The purpose (backup, coaxial, and extension), efficacy, and safety of the guide extension catheters were explored. The efficacy of the guide extension catheter was assessed based on the success of the procedures and safety was evaluated based on procedure-related complications.

### Results

Regarding the intended use, a guide extension catheter was used to strengthen the backup force of the guiding catheter in 52% of cases, extend the tip of the catheter in 38% of cases, and maintain the coaxiality of the guiding catheter in 10% of cases. Procedural success was achieved in 92.7% of 55 sessions and in 95.6% of 91 lesions. Complex lesions had a lower success rate than simple lesions (p = 0.04). Regarding safety concerns, complications were observed in 5 of 55 sessions (9.1%) and 6 of 91 lesions (6.6%). Only one case of pulmonary artery dissection using a guide extension catheter was reported. Except for this one case, extension catheter-related complications were not observed.

**Data Availability Statement:** All relevant data are within the paper and its Supporting Information files.

**Funding:** The authors received no specific funding for this work.

**Competing interests:** The authors have declared that no competing interests exist.

## Conclusions

A guide extension catheter can be used safely in BPA procedures with anatomically complex pulmonary artery branches and complex lesions by increasing backup support.

## Introduction

Chronic thromboembolic pulmonary hypertension (CTEPH) is a progressive disease in which organized thrombosis causes chronic obstruction of the pulmonary artery, resulting in increased pulmonary vascular resistance and pulmonary hypertension [1,2]. It has been reported that pulmonary endarterectomy (PEA) can reduce pulmonary arterial pressure, improve symptoms, and improve prognosis in patients with central CTEPH that is surgically accessible to the proximal thrombus [3–6]. Pulmonary vasodilators, such as soluble guanylate cyclase stimulants, selective prostacyclin receptor agonists, and subcutaneous treprostinil are indicated for patients with inoperable or residual pulmonary hypertension after PEA [7–9]. Recently, many reports have demonstrated the effectiveness of balloon pulmonary angioplasty (BPA) in patients with inoperable CTEPH and residual pulmonary hypertension after PEA [10–13].

Although advances in technical procedures have improved BPA success rates and safety [14–16], some problems still need to be resolved. In BPA procedures, because the pulmonary artery has multiple branches and is anatomically complicated by a large branch angle and tortuous vascular remodeling, it is sometimes difficult to engage a guiding catheter deeply to obtain adequate coaxiality and backup force. In addition, organized thrombotic lesions, such as web, subtotal, and total occlusion lesions [17] are too solid to allow for balloon catheters to pass through. Deep engagement or a large guiding catheter to enhance the backup force increases the risk of procedure-related complications such as vascular dissection and hemoptysis due to vascular damage. Consistent with the learning curve of the BPA procedure [13,18], a careful and skillful guiding catheter operation is essential for BPA success and safety.

The guide extension catheter is a rapid exchange catheter that has been developed to extend and complement the guiding catheter. The usefulness and safety of guide extension catheters have been reported in percutaneous coronary interventions (PCIs) [19–21]. Guide extension catheters are used to strengthen backup support and assist in the delivery of stents or balloons for the treatment of complex lesions, such as flexion, tortuous, or calcified lesions. To avoid vascular injury, the tip of the catheter is coated with silicon and has a flexible structure that allowed deep seating of the guide extension catheter. Recently, guide extension catheters have also been used in peripheral vascular interventions [22–24]. Guide extension catheters are expected to be effective for BPA, where it is difficult to obtain coaxiality and a powerful backup force. However, the effectiveness and safety of guide extension catheters for BPA treatment in patients with CTEPH are not well known. Therefore, this study aimed to explore the usefulness of a guide extension catheter in BPA.

## Methods

### Study design and population

This retrospective multicenter study was approved by the institutional review boards of the Kyoto Prefectural University of Medicine (ERB-C-2196) and the Omihachiman Community Medical Center (ERB-R3-44). From August 2016 to February 2022, 28 consecutive patients

with CTEPH who underwent BPA using a guide extension catheter at the two institutes were analyzed, accounting for a total of 91 lesions in 55 sessions.

## Collection of clinical data

We collected clinical data such as age, sex, World Health Organization functional class (WHO-FC), six-minute walking distance (6MWD), brain natriuretic peptide (BNP), and medications from patient electronic medical records at time of treatment. Hemodynamic characteristics were assessed using right heart catheterization before BPA, and right atrial pressure (RAP), pulmonary artery wedge pressure (PAWP), pulmonary artery pressure (PAP), and cardiac output (CO) with thermodilution measured. We also recorded procedure characteristics, including the treated site, lesion type, access site, guiding catheter type, and purpose of using a guide extension catheter. Each pulmonary lobe was counted as a treatment site. Based on a previous report, the lesion type was classified into five types: ring-like, web, subtotal, total occlusion, and tortuous [15,25].

## BPA procedures and guide extension catheter

All BPA procedures were performed via the femoral or jugular vein approach. An 8-Fr sheath was inserted into the vein, and a 6-Fr guiding sheath (ParentPlus, MEDIKIT, Tokyo, Japan or BRITE TIP, Cordis, FL, USA) was advanced to the main pulmonary artery through the 8-Fr sheath using a 0.035-inch wire (Radifocus Guide Wire M; Terumo, Tokyo, Japan). Pulmonary artery branches were selected using a 6-Fr guiding catheter (Profit; NIPRO, Osaka, Japan, or Mach1; Boston Scientific Corporation, MA, USA). Pulmonary angiography was performed manually using a half-contrast medium diluted with saline. The guide extension catheter was a rapid exchange catheter with a soft flexible 25 cm straight push-rod, which was developed to complement the role of the guiding catheter by extending the latter. A 6Fr guide extension catheter (Guideliner PV, which is approved for peripheral intervention in Japan; Teleflex, Tokyo, Japan; Fig 1) was used when it was difficult to complete BPA by general procedure; for example when the backup force of the guiding catheter was insufficient for crossing the < 2 mm balloon catheter to the target lesion (backup), when coaxiality between the guiding catheter and pulmonary artery was not matched and resulted in poor contrast image (coaxial), or when the guiding catheter could not reach the target vessel due to a large vessel angle (extension). In all procedures, the choice of device, including the guide extension catheter, was left to the discretion of the operators.

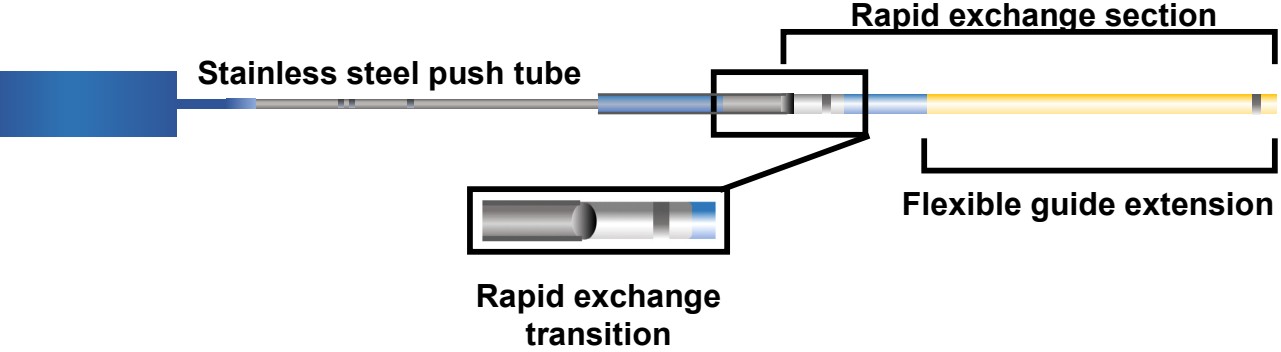

**Fig 1. Guide extension catheter.** The guide extension catheter is a rapid-exchange type catheter that provides backup support to deliver intervention devices such as balloons in difficult cases.

### Evaluation of effectiveness and safety of guide extension catheter

The efficacy of the guide extension catheter was assessed based on the success of the procedure, and its safety was evaluated based on procedure-related complications. Procedure success was defined as successful delivery and inflation of the balloon catheter to the target lesion using a guide extension catheter. Procedure-related complications related to the guide extension catheter were defined as complications in the BPA procedures of dissection, hemoptysis, and oozing.

### Statistical analysis

Continuous data are expressed as mean ± standard deviation (SD). Data that were not normally distributed are presented as medians (interquartile ranges). Categorical variables were compared using the Fisher's exact test. Statistical significance was set at $p < 0.05$. Statistical analyses were performed using the JMP software version 11.0.0 (SAS Institute Inc., Cary, NC, USA).

## Results

### Patients characteristics

Table 1 shows the baseline characteristics of the 28 patients. Mean age was 66.3 ± 12.5 years and 82.1% of patients were female. Six-minute walking distance (6MWD) was decreased to 376.2 ± 114.9 m. All patients had WHO-FC II or III with BNP levels of 95.2 pg/mL (23.4–342.4

**Table 1. Baseline population characteristics.**

|  | All patients (n = 28) |
| --- | --- |
| Age, years | 66.3±12.5 |
| Female, n (%) | 23 (82.1) |
| WHO-FC I/II/III/IV, n | 0/16/12/0 |
| 6MWD, m | 376.2±114.9 |
| BNP, pg/mL | 95.2 (23.4–342.4) |
| Medication, n (%) |  |
| sGC stimulator | 17 (61) |
| IP receptor agonist | 0 (0) |
| Diuretics | 14 (50) |
| Hemodynamics |  |
| RAP, mmHg | 6.8±3.7 |
| PAWP, mmHg | 9.3±2.9 |
| sPAP, mmHg | 66.0±15.7 |
| dPAP, mmHg | 22.0±6.6 |
| mPAP, mmHg | 37.4±9.0 |
| CO, L/min | 3.9±1.3 |
| CI, L/min/m$^2$ | 2.47±0.63 |
| PVR, WU | 8.2±4.4 |

Data are presented as n (%), mean ± SD or median (interquartile range). WHO-FC, World Health Organization functional class; 6MWD, 6-minute walk distance; BNP, brain natriuretic peptide; sGC, soluble guanylate cyclase; IP, prostacyclin; RAP, right atrial pressure; PAWP, pulmonary artery wedge pressure; sPAP, systolic pulmonary artery pressure; dPAP, diastolic pulmonary artery pressure; mPAP, mean pulmonary artery pressure; CO, cardiac output; CI, cardiac index; PVR, pulmonary vascular resistance.

pg/mL). Seventeen patients (61%) received soluble guanylate cyclase stimulators and no patients received prostacyclin receptor agonists. In terms of hemodynamics parameters, mean pulmonary artery pressure (mPAP) was $37.4 \pm 9.0$ mmHg and cardiac output (CO) was $3.9 \pm 1.3$ L/min. Mean right atrium pressure (RAP) was $6.8 \pm 3.7$ mmHg and pulmonary artery wedge pressure (PAWP) was $9.3 \pm 2.9$ mmHg; pulmonary vascular resistance (PVR) was calculated as $8.2 \pm 4.4$ wood units.

## Procedural characteristics

The procedural characteristics for the 91 lesions in 55 sessions are presented in Table 2. A guide extension catheter was used bilaterally in all areas of the pulmonary artery. Although the most common lesion type was a web lesion, a guide extension catheter was often used for complex subtotal, total occlusion, and tortuosity lesions. More than half of the cases (65%) were approached from the femoral vein and the remaining were approached from the jugular vein (35%). To guide catheter shapes, the JR4 guiding catheter was selected in many cases (56%). In complex lesions, the guide extension catheter was used most frequently to strengthen the

**Table 2. Procedure characteristics.**

| | All lesions (n = 91) |
|---|---|
| Total sessions, n | 55 |
| Juglar approach, n (%) | 18 (32.7) |
| Femoral approach, n (%) | 37 (67.3) |
| Treated site | |
| Right PA | |
| Upper lobe, n (%) | 13 (14) |
| Middle lobe, n (%) | 12 (13) |
| Lower lobe, n (%) | 18 (20) |
| Left PA | |
| Upper lobe, n (%) | 13 (14) |
| Lingular, n (%) | 19 (21) |
| Lower lobe, n (%) | 16 (18) |
| Lesion type | |
| Ring-like, n (%) | 3 (3) |
| Web, n (%) | 51 (56) |
| Subtotal, n (%) | 15 (17) |
| Total occlusion, n (%) | 11 (12) |
| Tortuous, n (%) | 11 (12) |
| Guiding catheter type | |
| JR4, n (%) | 51 (56) |
| JL4, n (%) | 5 (5) |
| AL1, n (%) | 16 (18) |
| MP, n (%) | 19 (21) |
| Purpose of guide extension | |
| Backup, n (%) | 47 (52) |
| Coaxial, n (%) | 9 (10) |
| Extension, n (%) | 35 (38)f |

Data are presented as n (%). PA, pulmonary artery; JR, Judkins Right; JL, Judkins Left; AL, Amplatz Left; MP, Multipurpose.

backup force of the guiding catheter (52%), and in 38% of cases were used to extend the tip of the catheter to engage in the pulmonary segmental artery with a large branching angle (Fig 2). The others were used to keep the guiding catheter coaxial to the pulmonary artery branches by advancing the guide extension in the direction of the pulmonary artery branch.

### Efficacy of guide extension catheter in BPA procedure

The procedural results are shown in Tables 3 and 4. Among 55 sessions, the guide extension catheter was used in 1.7 ± 0.8 lesions per session. Procedural success reached 92.7% (5 of 55 sessions). Complications were observed in 5 of 55 sessions (9.1%). Although we observed hemoptysis associated with BPA, there were no serious complications such as use of non-invasive positive pressure ventilation (NPPV) or mechanical ventilator. Regarding the lesion types, procedural success was achieved in 87 of 91 lesions (95.6%). Complex lesions had a lower success rate than simple ring-like and web lesions (ring-like: 100%, web lesion: 100%, subtotal: 93.3%, total occlusion: 81.8%, tortuous: 90.9%, p = 0.04). The details of the failed cases are presented in Table 5. Most of the failed lesions were complex subtotal, total occlusion, and tortuosity lesions, and the guide extension catheter was mainly used for backup reinforcement.

### Safety of guide extension catheter in BPA procedure

Regarding safety concerns, complications were observed in 6 of 91 lesions. Table 6 presents the details of these complications. Hemoptysis and oozing caused by the guidewire operation were observed in 2 lesions (2.2%) and 3 lesions (3.3%), respectively. There was only one case (1.1%) with pulmonary artery dissection using a guide extension catheter. Subtotal lesions had a significantly higher complication rate, particularly oozing by guidewire injuries, than other lesions (ring-like, 0%; web lesion, 2.0%; subtotal, 26.7%; total occlusion, 9.1%; tortuous, 0%; p = 0.02). In the case of dissections (Fig 3), the JR4.0 type guiding catheter was engaged to the right middle lobe branch. The web lesion was too tight to be crossed with a 1.5 mm balloon catheter. Therefore, a guide extension catheter was inserted deeply to strengthen the backup force. Using a guide extension catheter, a 1.5 mm balloon was allowed to pass through the lesion. However, pulmonary angiography after balloon dilatation revealed vascular dissection, which was suspected to be an extension-catheter-induced dissection. Bailout was possible by dilating the true lumen with a large-diameter balloon. Except for this one case, extension catheter-related complications were not observed.

## Discussion

Our findings in this study were as follows: guide extension catheters can be used to enhance backup force, to engage the target vessel, or to maintain the coaxiality of the guiding catheter against lesions. The guide extension catheter was effectively deployed in successful procedures, and most failed cases involved complex lesions. Complications were found in 6.6% of cases, but most of them occurred due to guidewire manipulation and device-related complications were seen in only one case (1.1%). Therefore, a guide extension catheter may be an effective and safe tool for BPA.

CTEPH is known to have a poor prognosis even in recent years, but it has been reported that BPA improves pulmonary hypertension, prevents the progression of right heart failure, and improves prognosis and quality of life [26–29]. Although BPA has rapidly developed in recent years, there are no established methods for this procedure. As the pulmonary artery has many segmental branches with complex anatomical morphology, the shape of the guiding catheter often does not match some segmental branches, and a sufficient backup force cannot be obtained [30]. The guide extension catheter is a rapid exchange-type catheter designed for

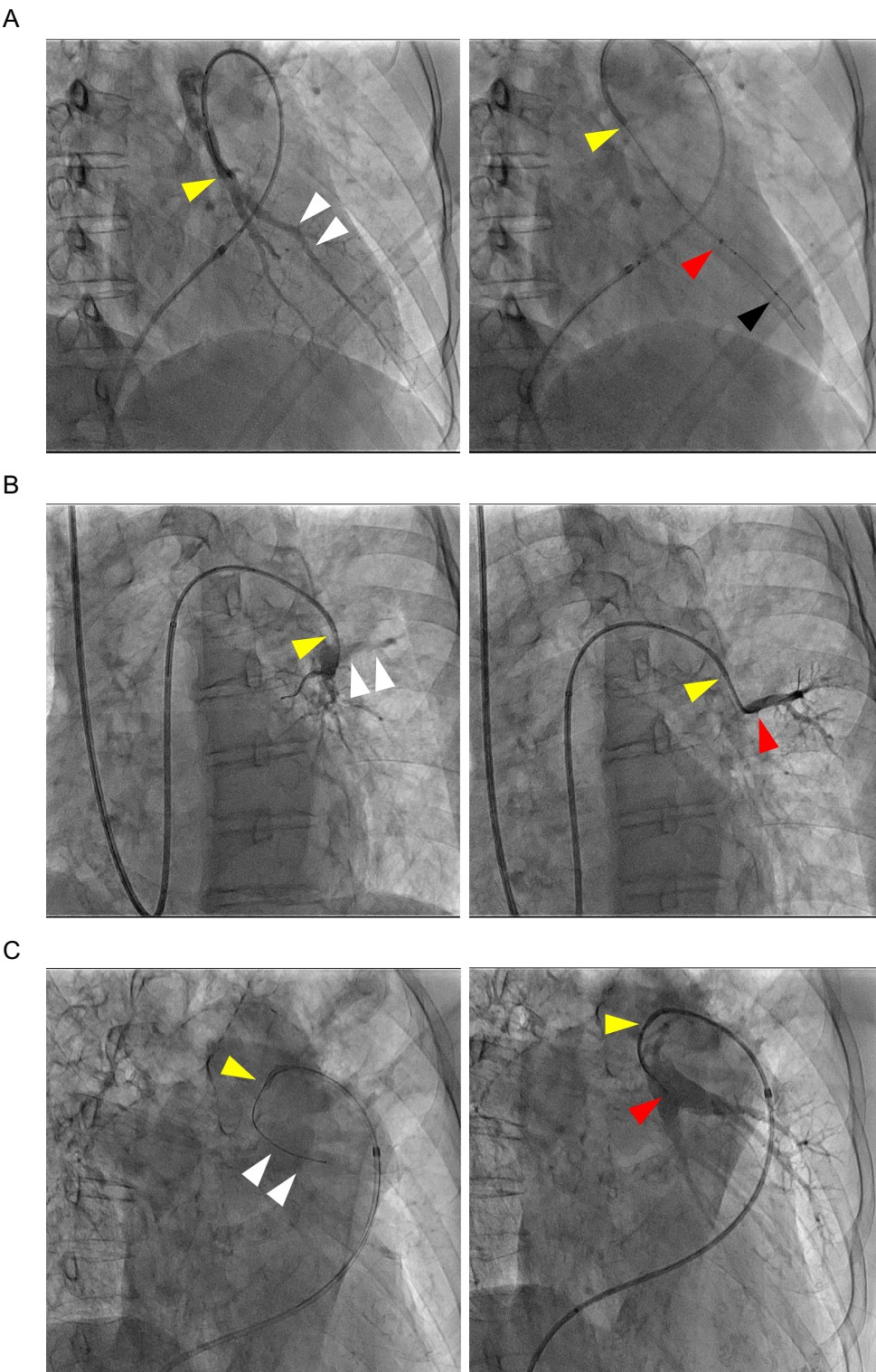

**Fig 2. Representative figures of guide extension catheter.** (A) A 6Fr Profit JR4 type guiding catheter was engaged to the left lower lobe branch (A8). 2 mm balloon catheter (IKAZUCHI PAD, KANEKA, Osaka, Japan) could not pass the web lesion and guide extension catheter was used for strengthening backup force. (B) Target vessel in back branch of left lower lobe (A8) had too large a branch angle to obtain coaxiality from the 6Fr Profit JR4 type guiding catheter, so the guide extension catheter was advanced to the target branch through the guidewire and balloon catheter. (C) Pulmonary artery was markedly enlarged and the 6Fr Profit JR4 type guiding catheter did not reach the lingular branch (A5). After a guidewire (B-pahm; Japan Lifeline, Tokyo, Japan) was inserted, we were able to selectively insert

the guide extension catheter using the slip-through technique with a 2 mm balloon catheter (IKAZUCHI PAD). The white arrowheads in the left panels of pulmonary angiography show target vessels. The yellow arrowheads show the tip of guiding catheter, and the red arrowheads represent the tip of the guide extension catheter. The black arrowhead is a marker of the balloon catheter.

**Table 3. Procedural results.**

|  | Total sessions (n = 55) |
|---|---|
| Lesions treated with guide extension catheter in a session, n (%) | 1.7 ± 0.8 |
| Procedural success, n (%) | 51 (92.7) |
| Complications, n (%) | 5 (9.1) |
| Hemoptysis, n (%) | 5 (9.1) |
| Usage of NPPV, n (%) | 0 (0) |
| Endotracheal intubation, n (%) | 0 (0) |

Data are presented as n (%) or mean ± SD. NPPV; non-invasive positive pressure ventilation.

**Table 4. Procedural results by lesion types.**

|  | Total (n = 91) | Ring-like (n = 3) | Web (n = 51) | Subtotal (n = 15) | Total occlusion (n = 11) | Tortuous (n = 11) | P value |
|---|---|---|---|---|---|---|---|
| Procedural success, n (%) | 87 (95.6) | 3 (100) | 51 (100) | 14 (93.3) | 9 (81.8) | 10 (90.9) | 0.04 |
| Complications, n (%) | 6 (6.6) | 0 | 1 (2.0) | 4 (26.7) | 1 (9.1) | 0 | 0.02 |
| Dissection, n (%) | 1 (1.1) | 0 | 1 (2.0) | 0 | 0 | 0 | 1.00 |
| Hemoptysis, n (%) | 2 (2.2) | 0 | 0 | 1 (6.7) | 1 (9.1) | 0 | 0.22 |
| Oozing, n (%) | 3 (3.3) | 0 | 0 | 3 (20.0) | 0 | 0 | 0.01 |

Data are presented as n (%).

**Table 5. Failed cases.**

| Case No. | Treated site | Lesion type | Access | Guiding | Purpose | Reasons for failure |
|---|---|---|---|---|---|---|
| 1 | Rt. A3 | Subtotal | Femoral | JR4 | Backup | Hemoptysis |
| 2 | Lt. A8 | Total | Juglar | MP | Backup | Wire could not cross |
| 3 | Rt. A10 | Total | Juglar | JR4 | Backup | Wire could not cross |
| 4 | Rt. A4 | Tortuous | Femoral | AL1 | Extension | Guide extension catheter could not cross |

JR, Judkins Right; MP, Multipurpose; AL, Amplatz Left; Rt, Right; Lt, Left.

**Table 6. Complications.**

| Case No. | Complication | Treated site | Lesion type | Access | Guiding | Purpose |
|---|---|---|---|---|---|---|
| 1 | Hemoptysis | Rt. A3 | Subtotal | Femoral | JR4 | Backup |
| 2 | Hemoptysis | Rt. A5 | Total | Femoral | JR4 | Backup |
| 3 | Oozing | Rt. A1 | Subtotal | Femoral | AL1 | Extension |
| 4 | Oozing | Rt. A4 | Subtotal | Femoral | JR4 | Backup |
| 5 | Oozing | Rt. A3 | Subtotal | Femoral | JR4 | Backup |
| 6 | Dissection | Rt. A5 | Web | Juglar | JR4 | Backup |

JR, Judkins Right; MP, Multipurpose; AL, Amplatz Left; Rt, Right; Lt, Left.

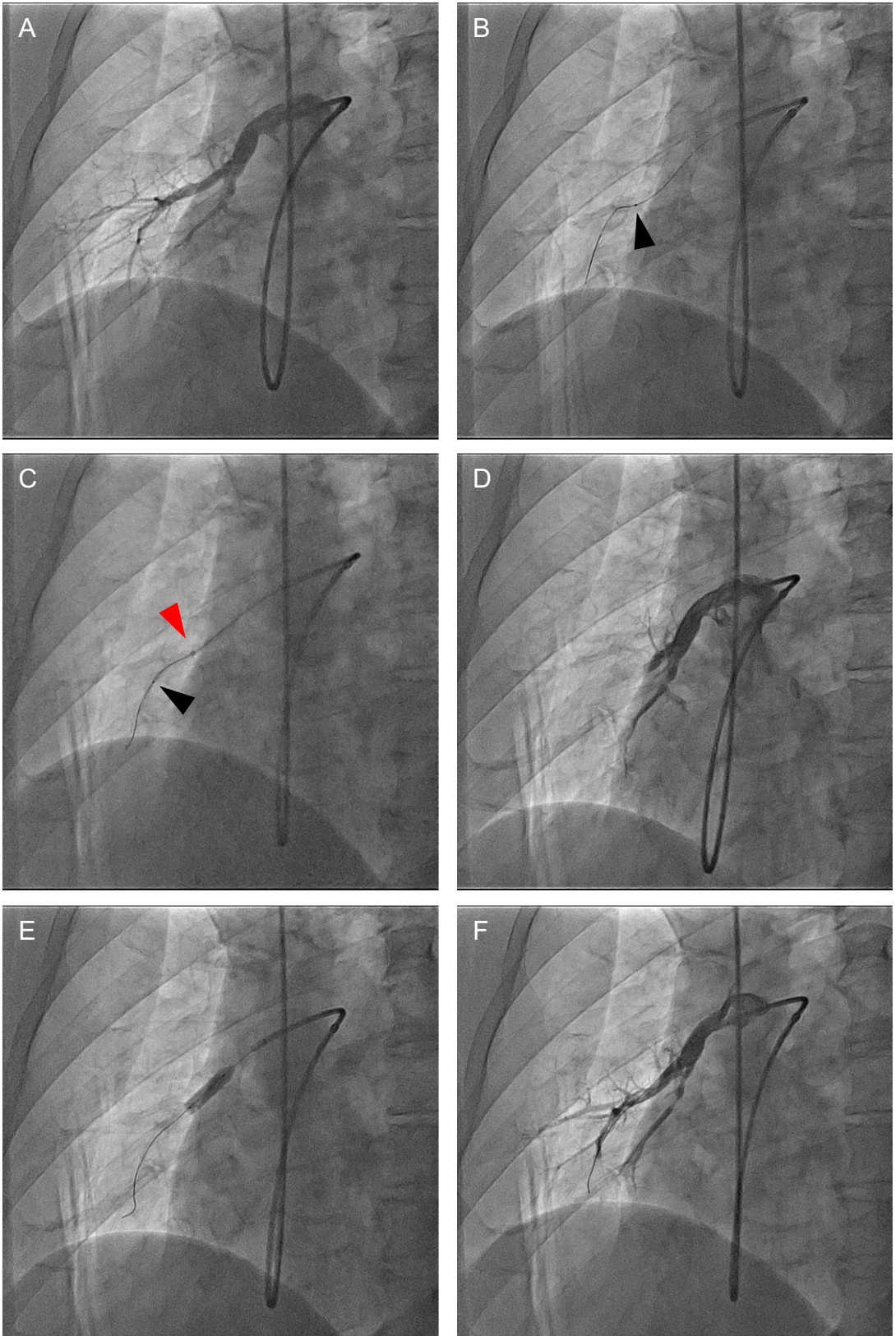

**Fig 3. Case of complication with vascular dissection by guide extension catheter.** (A) Pulmonary angiography at right middle lobe (A5) with a 6Fr Profit JR4 guiding catheter before BPA. (B) 1 mm Balloon catheter (IKAZUCHI Zero, KANEKA, Osaka, Japan) could not pass the web lesion (Black arrowhead indicates the tip of the balloon catheter). (C) After the guide extension catheter was advanced to strengthen backup force, the balloon catheter was able to cross the lesion (Red arrowhead represents the tip of the guide extension catheter). (D) Angiography after balloon dilatation revealed severe vascular dissection due to guide extension catheter. (E) After the guidewire (B-pahm 0.6 g) had passed through the true lumen confirmed with intravascular ultrasound (IVUS, Eagle Eye Platinum; Volcano, San Diego, CA), balloon dilation with large diameter balloon (4.0 mm IKAZUCHI PAD) was performed. (F) Final angiography demonstrated successful bailout with anterograde flow to the distal branches.

deep seating of a vessel. This can provide additional support and assist in device delivery. Guide extension catheters are widely used in PCI and their safety and effectiveness have been well established. The success rate of using guide extension catheters for complex PCIs such as calcification and chronic total occlusion lesions was 80–99% [20,21,31–35]. On the other hand, complications were very low at approximately 2%, and were mainly stent dislodgements, air embolisms, and vascular dissections [31,32,36]. Moreover, in recent years, guide extension catheters have also been used for various catheter interventions such as treatment of the renal and lower limb arteries [22–24].

In this study, guide extension catheters were used safely and effectively for complicated lesions in BPA procedures. In a previous report, the proportion of complex lesions excluding web and ring-like stenosis was 23% of all lesions treated with BPA [17]. However, in this study, 41% of the lesions were complex lesions that were hard and required backup force for penetration. Hence, the guide extension catheter was primarily used to strengthen the backup force of the guiding catheter. BPA is a procedure in which the guiding catheter floats in the large pulmonary artery and thus it is difficult to obtain sufficient backup force. A guide extension catheter can overcome this problem and is considered useful in BPA procedures.

Our overall success rate was very high (95.6%). In the case of simple lesions such as ring-like and web lesions, the success rate was 100%. Even in complex lesions such as total occlusion, the success rate was higher than that in previous reports [17]. These results demonstrate that the guide extension catheter is effective for BPA. In the unsuccessful cases, three were total occlusion or subtotal lesions in which it was difficult to cross the wire or balloon catheter to the hard lesions. Guide extension catheters have been used to enhance backup support; however, in harder lesions, the backup support from a guide extension catheter is limited. If a more powerful backup force is needed, the shape of the guiding catheter could possibly be changed to increase coaxiality to the lesion, or a large guiding catheter such as 7 Fr or 8 Fr guiding catheter can be used to increase the backup force. However, large guiding catheters increase risks of vascular injury, and therefore attention must be paid to careful management.

Complications occurred in six lesions, most of which were caused by guidewire operations. There was only one case of vascular dissection due to direct injury to the guide extension catheter. Because the pulmonary artery, which consists of venous tissue, is more vulnerable than the artery, deep insertion of a guide extension catheter should be performed carefully to prevent vessel dissection.

This study had several limitations. First, this study was conducted retrospectively at two institutes with a limited sample size and was not a comparative study. A randomized controlled trial with a larger number of patients is required to confirm the utility of guide extension catheters in BPA. Second, there were several biases in patient and procedure selection due to patient selection and use of the guide extension catheter were left to BPA operators. However, in many cases, a guide extension catheter was used because the lesion was tight, the devices could not pass, and effectiveness could be evaluated sufficiently. Third, because BPA is usually performed on many lesions in one session, we were unable to assess whether the use

of a guide extension catheter improved hemodynamics and contributed to improved prognosis.

## Conclusion

A guide extension catheter can be safely used during BPA procedures. It may be an optional tool to provide a guiding system that allows for more appropriate insertions into anatomically complex pulmonary artery branches or successfully to provide additional backup support to approach hard complex lesions.

## Supporting information

**S1 Checklist. STROBE statement—checklist of items that should be included in reports of observational studies.**
(DOCX)

**S1 File.**
(XLSX)

## Acknowledgments

We would like to acknowledge Editage (www.editage.com) for providing assistance with English language editing.

## Author Contributions

**Conceptualization:** Masao Takigami, Naohiko Nakanishi, Hideo Tsubata.

**Investigation:** Masao Takigami, Hideo Tsubata, Kuniyoshi Fukai, Yuki Matsubara.

**Project administration:** Naohiko Nakanishi.

**Resources:** Masao Takigami, Hideo Tsubata, Kuniyoshi Fukai, Yuki Matsubara, Kenji Yanishi, Kan Zen, Takeshi Nakamura.

**Supervision:** Satoaki Matoba.

**Writing – original draft:** Masao Takigami, Naohiko Nakanishi.

**Writing – review & editing:** Hideo Tsubata, Kuniyoshi Fukai, Yuki Matsubara, Kenji Yanishi, Kan Zen, Takeshi Nakamura, Satoaki Matoba.

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
