## [Decision Letter · Decision Letter 0]

6 Dec 2022

PONE-D-22-27862

Efficacy and safety of guide extension catheter in balloon pulmonary angioplasty for treatment of complex lesions in chronic thromboembolic pulmonary hypertension

PLOS ONE

Dear Dr. Nakanishi,

Thank you for submitting your manuscript to PLOS ONE. After careful consideration, we feel that it has merit but does not fully meet PLOS ONE’s publication criteria as it currently stands. Therefore, we invite you to submit a revised version of the manuscript that addresses the points raised during the review process.

We look forward to receiving your revised manuscript.

Kind regards,

Redoy Ranjan, MBBS, MRCSEd, Ch.M., MS (CV&TS), FACS

Academic Editor

PLOS ONE

Journal Requirements:

Reviewers' comments:

Reviewer's Responses to Questions

**Comments to the Author**

1. Is the manuscript technically sound, and do the data support the conclusions?

Reviewer #1: Yes

Reviewer #2: Partly

Reviewer #3: Yes

Reviewer #4: Partly

Reviewer #5: Yes

2. Has the statistical analysis been performed appropriately and rigorously? 

Reviewer #1: Yes

Reviewer #2: Yes

Reviewer #3: Yes

Reviewer #4: I Don't Know

Reviewer #5: Yes

3. Have the authors made all data underlying the findings in their manuscript fully available?

Reviewer #1: Yes

Reviewer #2: Yes

Reviewer #3: Yes

Reviewer #4: No

Reviewer #5: No

4. Is the manuscript presented in an intelligible fashion and written in standard English?

Reviewer #1: Yes

Reviewer #2: Yes

Reviewer #3: Yes

Reviewer #4: Yes

Reviewer #5: Yes

5. Review Comments to the Author

Reviewer #1: A retrospective analysis was performed on 91 cases of 28 patients with chronic thromboembolic pulmonary hypertension. All patients underwent balloon pulmonary angioplasty using guided extension catheter. The results suggest that guided extension catheters can be used effectively and safely for successful surgery by adding backup support to anatomically complex pulmonary artery branches and complex lesions during BPA surgery.

Reviewer #2: The authors present data on use of guide extension, lesion characteristics and relate it to procedural success / complications in BPA for CTEPH. There are limited data on optimal procedural technique and practice is varied. The lesion type success/ complication data is of interest but the main findings focus on guide extension data that is subjective e.g. coaxiality etc and of limited interest and use to operators. Whilst of some reassurance this paper is not practice changing.

Reviewer #3: The authors retrospectively analyzed 91 lesions in 28 patients with CTEPH who underwent BPA using a guide extension catheter, aiming to assess efficacy and safety of guide extension catheter in balloon pulmonary angioplasty for treatment of complex lesions in chronic thromboembolic pulmonary hypertension.

This is a cross-sectional descriptive study. The rational and method of the study were clearly stated and results were original and presented in sufficient detail. The basic and standard statistical analysis were performed. The conclusion is reasonable. The study was limited by its small sample size, study design and selection bias. All these were thoroughly discussed by the authors.

There are a few minor issues need to be addressed:

1. Please read the manuscript carefully to corrected a few grammar errors.

2. I think fisher’s exact test is needed to compare some of the categorical variables, since some of the cells are less than 5.

Reviewer #4: Nakanishi et al describe their experience using guide extensions and report that these are safe and effective in BPA.

While it may seem obvious that having a proven track record in other vascular systems, guide extensions would also work in the pulmonary vasculature, it is reasonable to provide some published evidence to back this belief.

The work is retrospective. It is not clearly stated whether any consent for using patient data was obtained, but as an observational audit of standard practice this is less important. However, this leads to uncertainty about the indications for use since there is no evidence that such information was collected prospectively. It is not clear that a failure to cross each lesion was required before the guide extension was used, nor how the indication was recorded for each lesion treated. Given that the extension appears to have been used in nearly 4 lesions per patient, it may be that once available it was used for all lesions treated at that session. It is also possible that it was only used once in each of these patients per session – but that multiple sessions per patient are being reported.

It would help put the work in perspective if we were given data on the total number of patients, sessions and lesions treated by the team over the study period, and similar data about guide extension usage. Without such data it is unclear that usage was clinically necessary rather than just operator preference.

The complication rate is acceptable, one must wonder whether the catheter induced dissection related to contrast injection through a damped catheter. If so, this is simple operator error rather than a specific catheter related complication.

The ‘failure’ rate in web lesions seem remarkably high – perhaps providing background data would help put this in perspective.

From Figure 2c – it would appear that the only purpose of the guide extension was to facilitate imaging of the lingular branch, as it is implied that the guidewire and balloon crossed the lesion before the extension was advanced using a slip through technique.

The precise definitions of lesion morphology are not provided. For example was the modified Kawakami classification used?

I would conclude that it is worth publishing the experience as evidence of safety. Little can be said of efficacy or applicability with the information provided.

Reviewer #5: MAJOR POINTS:

1. The Guideliner was used in this study. Authors should comment on the use of other Guide Extension Catheters, e.g., the Medtronic Telescope

2. Were more guide extension catheters used for taller patients?

3. Authors should indicate how often the rapid exchange transition could not be crossed easily with a second wire.

4. In the Figures, angiographic projections need to be described. Furthermore, segment names should be provided, as well as balloon sizes, wires and their brandings.

5. Complications should also be reported by session to allow for comparison with published data, and not only by lesion.

6. If complications were reported by lesion, were those only lesions addressed with the Guideliner, or all lesions. Please, clarify.

MINOR POINTS :

7. Please correct: Pulmonary vasodilators, soluble guanylate cyclase stimulants and selective prostacyclin receptor agonists, and SC Treprostinil are indicated for patients with inoperable or residual pulmonary hypertension.

8. …. and improves prognosis and quality of life [24– 270 27]. Please choose more recent manuscripts for these statements

6. PLOS authors have the option to publish the peer review history of their article (what does this mean?). If published, this will include your full peer review and any attached files.

Reviewer #1: No

Reviewer #2: No

Reviewer #3: **Yes: **Wanzhu Zhang

Reviewer #4: **Yes: **J Gerry Coghlan

Reviewer #5: **Yes: **Irene Lang

---

## [Author Response · Author response to Decision Letter 0]

28 Dec 2022

We thank the Editor and Reviewers for their comments and suggestions, which we have responded to. We hope that we have adequately modified the revised manuscript to address all their concerns. Changes made in response to all queries are indicated using the track changes feature in the revised manuscript.

 

Reviewer #1

A retrospective analysis was performed on 91 cases of 28 patients with chronic thromboembolic pulmonary hypertension. All patients underwent balloon pulmonary angioplasty using guided extension catheter. The results suggest that guided extension catheters can be used effectively and safely for successful surgery by adding backup support to anatomically complex pulmonary artery branches and complex lesions during BPA surgery.

Response:

Thank you for your insightful comment. As you commented, our data supports the safety and efficacy of guide extension catheter in BPA. Thus, guide extension catheters can be an optional device for complex lesions.

 

Reviewer #2

The authors present data on use of guide extension, lesion characteristics and relate it to procedural success / complications in BPA for CTEPH. There are limited data on optimal procedural technique and practice is varied. The lesion type success/ complication data is of interest but the main findings focus on guide extension data that is subjective e.g. coaxiality etc and of limited interest and use to operators. Whilst of some reassurance this paper is not practice changing.

Response:

We very much appreciate your valuable comments and suggestions. As you mentioned, this study has many limitations and biases, making it difficult to fully prove the efficacy and safety of guide extension.

In this study, the guide extension catheter worked effectively on 91 lesions of various types in 55 sessions. It was used to complement the role of the guiding catheter when the latter was found inadequate. More specifically, a guide extension catheter was used when it was difficult to complete BPA by general procedure; for example, in the case that we could not cross the balloon catheter with less than 2 mm due to insufficient backup force, that we could not obtain good contrast image due to axis misalignment, and that the guiding catheter could not reach the target vessel due to a large vessel angle. Among 55 sessions, the guide extension catheter was used in 1.7 ± 0.8 lesions per session. Procedural success reached 92.7 % (51 of 55 sessions). Complications were observed in 5 of 55 sessions (9.1 %). Although we observed hemoptysis associated with BPA, there were no serious complications such as use of non-invasive positive pressure ventilation (NPPV) or mechanical ventilator.

 We were unable to perform a statistical comparison of the efficacy of the extension catheters; however, our experience has demonstrated acceptable safety in the use of extension catheters. We believe that this data will help show the usefulness of guide extension catheter as an optional device for complex lesions in BPA, as it has demonstrated a high success rate with few complications despite being used for lesions that could not be easily treated with general BPA strategy.

 

Reviewer #3

The authors retrospectively analyzed 91 lesions in 28 patients with CTEPH who underwent BPA using a guide extension catheter, aiming to assess efficacy and safety of guide extension catheter in balloon pulmonary angioplasty for treatment of complex lesions in chronic thromboembolic pulmonary hypertension.

This is a cross-sectional descriptive study. The rational and method of the study were clearly stated and results were original and presented in sufficient detail. The basic and standard statistical analysis were performed. The conclusion is reasonable. The study was limited by its small sample size, study design and selection bias. All these were thoroughly discussed by the authors.

There are a few minor issues need to be addressed:

1. Please read the manuscript carefully to corrected a few grammar errors.

Response:

We greatly appreciate your suggestion. We had the help of Editage (www.editage.com) for the English language editing in the revised manuscript and have attached the editing certificate.

2. I think fisher’s exact test is needed to compare some of the categorical variables, since some of the cells are less than 5.

Response:

As you suggested, we have used Fisher’s exact test for the comparison of the categorical variables in Table 4 (newly added). 

Reviewer #4

Nakanishi et al describe their experience using guide extensions and report that these are safe and effective in BPA.

While it may seem obvious that having a proven track record in other vascular systems, guide extensions would also work in the pulmonary vasculature, it is reasonable to provide some published evidence to back this belief.

The work is retrospective. It is not clearly stated whether any consent for using patient data was obtained, but as an observational audit of standard practice this is less important. However, this leads to uncertainty about the indications for use since there is no evidence that such information was collected prospectively. It is not clear that a failure to cross each lesion was required before the guide extension was used, nor how the indication was recorded for each lesion treated. 

Response:

We greatly appreciate your comment. In this study, a guide extension catheter was used to complement the role of the guiding catheter when it was found inadequate. More specifically, a guide extension catheter was used when it was difficult to complete BPA by general procedure; for example, in the case that we could not cross the balloon catheter with less than 2 mm due to insufficient backup force, that we could not obtain good contrast image due to axis misalignment, and that the guiding catheter could not reach the target vessel due to a large vessel angle. The manuscript has been revised as follows to further clarify its usage criteria.

(Method in the revised manuscript, p9-10, line 128-134)

A 6Fr guide extension catheter (Guideliner PV which is approved for peripheral intervention in Japan; Teleflex, Tokyo, Japan; Figure 1) was used when it was difficult to complete BPA by general procedure; for example, when the backup force of the guiding catheter was insufficient for crossing the < 2 mm balloon catheter to the target lesion (backup), when coaxiality between the guiding catheter and pulmonary artery was not matched and resulted in poor contrast image (coaxial), or when the guiding catheter could not reach the target vessel due to a large vessel angle (extension).

Given that the extension appears to have been used in nearly 4 lesions per patient, it may be that once available it was used for all lesions treated at that session. It is also possible that it was only used once in each of these patients per session – but that multiple sessions per patient are being reported.

It would help put the work in perspective if we were given data on the total number of patients, sessions and lesions treated by the team over the study period, and similar data about guide extension usage. Without such data it is unclear that usage was clinically necessary rather than just operator preference.

Response:

Thank you for pointing out this important issue. Our paper might mislead the readers into thinking that the guide extension catheter may have been used for many lesions per a single session. We apologize for the confusion. To clarify this point, we have presented the session data (Table 3). These have been added to the manuscript as follows:

(Result in the revised manuscript, p16, line 219-224)

The procedural results are shown in Table 3 and Table 4. Among 55 sessions, the guide extension catheter was used in 1.7 ± 0.8 lesions per session. Procedural success reached 92.7 % (51 of 55 sessions). Complications were observed in 5 of 55 sessions (9.1 %). Although we observed hemoptysis associated with BPA, there were no serious complications such as use of non-invasive positive pressure ventilation (NPPV) or mechanical ventilator. 

(Result in the revised manuscript, p17, line 235-237)

Table 3. Procedural results

 Total sessions

(n=55)

Lesions treated with guide extension catheter in a session, n (%) 1.7 ± 0.8

Procedural success, n (%) 51 (92.7)

Complications, n (%) 5 (9.1)

Hemoptysis, n (%) 5 (9.1)

Usage of NPPV, n (%) 0 (0)

Endotracheal intubation, n (%) 0 (0)

Data are presented as n (%) or mean ± SD. NPPV; non-invasive positive pressure ventilation.

The complication rate is acceptable, one must wonder whether the catheter induced dissection related to contrast injection through a damped catheter. If so, this is simple operator error rather than a specific catheter related complication.

Response:

We completely agree with your point. We constantly checked for catheter dumps during the procedure. Contrast was injected after the guide extension catheter was retracted from the lesion in this session. Therefore, we assumed that the insertion of the guide extension catheter resulted in pulmonary artery branch injury.

The ‘failure’ rate in web lesions seem remarkably high – perhaps providing background data would help put this in perspective.

Response:

We greatly appreciate your comment. In our study, all procedures for web lesions have been completed successfully using a guide extension catheter and we experienced only one complication (dissection case). The four failure cases were complex subtotal, total occlusion, and tortuosity lesions, and these lesions had a high complication rate.

From Figure 2c – it would appear that the only purpose of the guide extension was to facilitate imaging of the lingular branch, as it is implied that the guidewire and balloon crossed the lesion before the extension was advanced using a slip through technique.

Response:

In the case of Figure 2c, a JR4 guiding catheter could not reach the lingular branch due to marked pulmonary artery dilation. Fortunately, the guide wire entered the lingular branch (not cross the lesion), but when the guiding catheter was pushed, it fell into the lower lobe branch and could not engage in the lingular branch. In this situation, in addition to not achieving sufficient contrast imaging, it was also difficult to cross the balloon catheter to the lesion because of poor backup force. The guiding extension catheter is flexible and can be extended toward the branch (‘Extension’) using the slip through technique.

We have revised Figure 2c to make it easier to see that the guide extension catheter is inserted using the slip through technique.

The precise definitions of lesion morphology are not provided. For example was the modified Kawakami classification used?

Response:

We had classified the lesions based on the Kawakami classification. This is described in the Method section (in the revised manuscript P9, line 113-114) citing the following references. We have added the following references regarding lesion morphology definition.

15. Kataoka M, Inami T, Kawakami T, Fukuda K, Satoh T. Balloon pulmonary angioplasty (percutaneous transluminal pulmonary angioplasty) for chronic thromboembolic pulmonary hypertension: A Japanese perspective. JACC Cardiovasc Intv. 2019; 12:1382-1388. doi: 10.1016/j.jcin.2019.01.237, PubMed PMID: 31103538.

25. Kawakami T, Ogawa A, Miyaji K, Mizoguchi H, Shimokawahara H, Naito T, et al. Novel Angiographic Classification of Each Vascular Lesion in Chronic Thromboembolic Pulmonary Hypertension Based on Selective Angiogram and Results of Balloon Pulmonary Angioplasty. Circ Cardiovasc Interv. 2016;9(10). doi: 10.1161/CIRCINTERVENTIONS.115.003318. PubMed PMID: 27729418.

I would conclude that it is worth publishing the experience as evidence of safety. Little can be said of efficacy or applicability with the information provided.

Response:

Thank you for your constructive comment. Safety is the most important issue in BPA. In this respect, we believe that our experience has demonstrated acceptable safety in the use of extension catheters. On the other hand, we were unable to perform a statistical comparison; thus, as you have pointed out, it is impossible to conclude the efficacy of the guide extension catheters based on our study alone. However, we believe that the guide extension catheter can be used as an option in BPA therapy because it has demonstrated a high success rate despite being an alternative strategy for lesions that cannot be easily treated with general BPA strategy. We have revised our manuscript to emphasize the safety of guide extension catheters in the conclusion. 

Reviewer #5

 MAJOR POINTS:

1. The Guideliner was used in this study. Authors should comment on the use of other Guide Extension Catheters, e.g., the Medtronic Telescope

Response:

Thank you for your comment. We used the Guideliner PV guide extension catheter because it is the only guide extension catheter approved for peripheral intervention in Japan. As you mentioned, other guide extension catheter can also be used as well. We have added this information in the Methods section as follow:

(Methods in the revised manuscript, p9, line 128)

Guideliner PV, which is approved for peripheral intervention in Japan

2. Were more guide extension catheters used for taller patients?

Response:

Unfortunately, since we did not compare patients with and without guide extension catheter in our study, we were unable to investigate the correlation between height and guide extension catheter use. In my opinion, patients requiring guided extension catheters may correlate with disease severity, not height. In patients with severe CTEPH, the lesions may be tough and difficult to cross a balloon catheter to them, or it may be difficult to maintain coaxiality or to engage a guiding catheter into a pulmonary branch due to marked pulmonary artery dilatation. We speculate that guide extension catheters can be used effectively in such situations.

3. Authors should indicate how often the rapid exchange transition could not be crossed easily with a second wire.

Response:

We performed BPA to 91 lesions in 55 sessions using a guide extension catheter. In this study, the second wire was not used to simplify the procedure. As you pointed out, if the second wire is inserted blindly in a guide extension catheter, it may sometimes be stuck to the rapid exchange transition of the guide extension catheter. Therefore, it is recommended to insert the second wire carefully under fluoroscopy guidance, or to use a microcatheter or a dual lumen catheter.

4. In the Figures, angiographic projections need to be described. Furthermore, segment names should be provided, as well as balloon sizes, wires and their brandings.

Response:

Thank you for pointing out this important issue. As you suggested, we have included names of target branches and the devices in the Figure legends as follows:

(Result Figure 2 in the revised manuscript, p15, line 202-212)

(A) A 6Fr Profit JR4 type guiding catheter was engaged to the left lower lobe branch (A8). 2 mm balloon catheter (IKAZUCHI PAD, KANEKA, Osaka, Japan) could not pass the web lesion and guide extension catheter was used for strengthening backup force. (B) Target vessel in back branch of left lower lobe (A8) had too large a branch angle to obtain coaxiality from the 6Fr Profit JR4 type guiding catheter, so the guide extension catheter was advanced to the target branch through the guidewire and balloon catheter. (C) Pulmonary artery was markedly enlarged and the 6Fr Profit JR4 type guiding catheter did not reach the lingular branch (A5). After a guidewire (B-pahm; Japan Lifeline, Tokyo, Japan) was inserted, we were able to selectively insert the guide extension catheter using the slip-through technique with a 2 mm balloon catheter (IKAZUCHI PAD). 

(Result Figure 3 in the revised manuscript, p19-20, line 269-280)

(A) Pulmonary angiography at right middle lobe (A5) with a 6Fr Profit JR4 guiding catheter before BPA. (B) 1 mm Balloon catheter (IKAZUCHI Zero, KANEKA, Osaka, Japan) could not pass the web lesion (Black arrowhead indicates the tip of the balloon catheter). (C) After the guide extension catheter was advanced to strengthen backup force, the balloon catheter was able to cross the lesion (Red arrowhead represents the tip of the guide extension catheter). (D) Angiography after balloon dilatation revealed severe vascular dissection due to guide extension catheter. (E) After the guidewire (B-pahm 0.6g) had passed through the true lumen, confirmed with intravascular ultrasound (IVUS, Eagle Eye Platinum; Volcano, San Diego, CA), balloon dilation with large diameter balloon (4.0 mm IKAZUCHI PAD) was performed. (F) Final angiography demonstrated successful bailout with anterograde flow to the distal branches.

5. Complications should also be reported by session to allow for comparison with published data, and not only by lesion.

Response:

Thank you for pointing out this important issue. Our paper might mislead readers to think that the guide extension catheter may have been used for many lesions per a single session. We apologize for the confusion. To clarify this point, we have presented the session data (Table 3). These have been added to the manuscript as follows:

(Result in the revised manuscript, p16, line 219-224)

The procedural results are shown in Table 3 and Table 4. Among 55 sessions, the guide extension catheter was used in 1.7 ± 0.8 lesions per session. Procedural success reached 92.7 % (51 of 55 sessions). Complications were observed in 5 of 55 sessions (9.1 %). Although we observed hemoptysis associated with BPA, there were no serious complications such as use of non-invasive positive pressure ventilation (NPPV) or mechanical ventilator. 

(Result in the revised manuscript, p17, line 235-237

Table 3. Procedural results

 Total sessions

(n=55)

Lesions treated with guide extension catheter in a session, n (%) 1.7 ± 0.8

Procedural success, n (%) 51 (92.7)

Complications, n (%) 5 (9.1)

Hemoptysis, n (%) 5 (9.1)

Usage of NPPV, n (%) 0 (0)

Endotracheal intubation, n (%) 0 (0)

Data are presented as n (%) or mean ± SD. NPPV; non-invasive positive pressure ventilation.

6. If complications were reported by lesion, were those only lesions addressed with the Guideliner, or all lesions. Please, clarify.

Response:

Thank you for your comment. Complications were limited to only those associated with the Guideliner in this manuscript. We have clarified this in the revised manuscript as follows. Complications by session have been described in Table 3 (newly added).

(Methods in the revised manuscript, p11, line 148)

Procedure-related complications related to the guide extension catheter were defined as complications in the BPA procedures of dissection, hemoptysis, and oozing.

MINOR POINTS :

7. Please correct: Pulmonary vasodilators, soluble guanylate cyclase stimulants and selective prostacyclin receptor agonists, and SC Treprostinil are indicated for patients with inoperable or residual pulmonary hypertension.

Response:

We thank you for pointing out this error. We apologize for not mentioning SC Treprostinil for CTEPH because it has not yet been approved in Japan. We have modified the introduction as below:

(Introduction in the revised manuscript, p5, line 60-62)

Pulmonary vasodilators, such as soluble guanylate cyclase stimulants, selective prostacyclin receptor agonists, and subcutaneous treprostinil are indicated for patients with inoperable or residual pulmonary hypertension [7–9].

9. Sadushi-Kolici R, Jansa P, Kopec G, Torbicki A, Skoro-Sajer N, Campean IA, et al. Subcutaneous treprostinil for the treatment of severe non-operable chronic thromboembolic pulmonary hypertension (CTREPH): a double-blind, phase 3, randomised controlled trial. Lancet Respir Med. 2019;7(3):239-48. doi: 10.1016/S2213-2600(18)30367-9. PubMed PMID: 30477763.

8. …. and improves prognosis and quality of life [24– 270 27]. Please choose more recent manuscripts for these statements

Response:

We thank you for your suggestion. Reference 25 has been replaced with a newer reference as given below:

(Reference in the revised manuscript, p33, line 488-492)

27. Hoole SP, Coghlan JG, Cannon JE, Taboada D, Toshner M, Sheares K, et al. Balloon pulmonary angioplasty for inoperable chronic thromboembolic pulmonary hypertension: the UK experience. Open Heart. 2020;7(1):e001144. doi: 10.1136/openhrt-2019-001144. PubMed PMID: 32180986; PubMed Central PMCID: PMCPMC7046957.

---

## [Decision Letter · Decision Letter 1]

6 Jan 2023

Efficacy and safety of guide extension catheter in balloon pulmonary angioplasty for treatment of complex lesions in chronic thromboembolic pulmonary hypertension

PONE-D-22-27862R1

Dear Dr. Nakanishi,

We’re pleased to inform you that your manuscript has been judged scientifically suitable for publication and will be formally accepted for publication once it meets all outstanding technical requirements.

Kind regards,

Redoy Ranjan, MBBS, MRCSEd, Ch.M., MS (CV&TS), FACS

Academic Editor

PLOS ONE

---

## [Editor Report · Acceptance letter]

20 Jan 2023

PONE-D-22-27862R1 

Efficacy and safety of guide extension catheter in balloon pulmonary angioplasty for treatment of complex lesions in chronic thromboembolic pulmonary hypertension 

Dear Dr. Nakanishi:

I'm pleased to inform you that your manuscript has been deemed suitable for publication in PLOS ONE. Congratulations! Your manuscript is now with our production department. 

Kind regards, 

on behalf of

Dr. Redoy Ranjan 

Academic Editor

PLOS ONE